 eLife

# Elevation of CpG frequencies in influenza A genome attenuates pathogenicity but enhances host response to infection

Eleanor Gaunt[1†], Helen M Wise[1†‡], Huayu Zhang[1], Lian N Lee[2], Nicky J Atkinson[1], Marlynne Quigg Nicol[1], Andrew J Highton[2], Paul Klenerman[2], Philippa M Beard[1], Bernadette M Dutia[1], Paul Digard[1], Peter Simmonds[1,2]*

[1]Infection and Immunity Division, Roslin Institute, University of Edinburgh, Edinburgh, United Kingdom; [2]Nuffield Department of Medicine, University of Oxford, Oxford, United Kingdom

*For correspondence: Peter.Simmonds@ed.ac.uk

†These authors contributed equally to this work

Present address: ‡Department of Engineering and Physical Sciences, Heriot Watt University, Edinburgh, United Kingdom

Competing interests: The authors declare that no competing interests exist.

**Abstract** Previously, we demonstrated that frequencies of CpG and UpA dinucleotides profoundly influence the replication ability of echovirus 7 (Tulloch *et al.*, 2014). Here, we show that that influenza A virus (IAV) with maximised frequencies of these dinucleotides in segment 5 showed comparable attenuation in cell culture compared to unmodified virus and a permuted control (CDLR). Attenuation was also manifested in vivo, with 10-100 fold reduced viral loads in lungs of mice infected with 200PFU of CpG-high and UpA-high mutants. However, both induced powerful inflammatory cytokine and adaptive (T cell and neutralising antibody) responses disproportionate to their replication. CpG-high infected mice also showed markedly reduced clinical severity, minimal weight loss and reduced immmunopathology in lung, yet sterilising immunity to lethal dose WT challenge was achieved after low dose (20PFU) pre-immunisation with this mutant. Increasing CpG dinucleotide frequencies represents a generic and potentially highly effective method for generating safe, highly immunoreactive vaccines.

## Introduction

The newly developed DNA technology to create viruses (*Wimmer et al., 2009*), bacteria (*Gibson et al., 2010*) and recently, whole eukaryotic chromosomes (*Annaluru et al., 2014*) with entirely synthetic genomes provides unparalleled opportunities to functionally explore replication and gene regulation. Insertion of sequences with large scale compositional changes using reverse genetic systems established for many RNA and DNA viruses has provided the opportunity to investigate the effect of virus genome composition on replication ability, evolutionary fitness and transmissibility. Large scale recoding of RNA virus genomes can be functionally exploited, for example to attenuate replication in a controllable way so as to create safer, non-reverting viral vaccines. Attenuation may be achieved by increasing the relative frequencies of disfavoured or rare codons encoding the same amino acid (*Bennetzen and Hall, 1982*; *Sharp et al., 2005*; *Wu et al., 2010*). Based on prokaryotic models, these impair translation rates and accuracy through mismatches with tRNA abundances and effects on ribosomal processivity (*Gingold and Pilpel, 2011*). Production of synthetic coding sequences with increasing frequencies of codons or codon pairs that are naturally under-represented in vertebrate and viral genomes have been similarly proposed to reduce translation efficiency (*Tats et al., 2008*; *Gutman and Hatfield, 1989*; *Yarus and Folley, 1985*; *Boycheva et al., 2003*; *Moura et al., 2005*; *Yang et al., 2013*; *Nogales et al., 2014*; *Fan et al., 2015*).

Current methods for virus attenuation have concentrated on sequence alterations designed to influence translation rates. However, it is increasingly recognised that both codon and codon pair de-optimisation (CPD) strategies have additional confounding effects on virus genome composition that are known to also influence virus replication (*Burns et al., 2009*; *Tulloch et al., 2014*; *Atkinson et al., 2014*). Significantly, selection for disfavoured coding elements increases frequencies of CpG and UpA dinucleotides that are normally suppressed in vertebrate RNA and DNA virus genomes. For RNA viruses, adding CpG and UpA dinucleotides in only a small part of a viral genome severely restricts virus replication, an effect that we have demonstrated to be independent of translation effects on the virus (*Tulloch et al., 2014*). The mechanism(s) through which dinucleotide composition influences virus replication rates are uncharacterised, and existing data indicates that they are independent of interferon pathways and PKR-mediated recognition of viral RNA (*Atkinson et al., 2014*).

In the current study, we have investigated whether the enhanced reactivity of cells to virus infection with high CpG/UpA frequencies in cell culture also modulates their replication and host immune responses in vivo. As an experimental model, we chose to use the A/Puerto Rico/8/34 (PR8) strain of influenza A virus (IAV) for which a reverse genetics system has been developed (*de Wit et al., 2004*) and which replicates in a variety of cell lines and causes pathogenic infections in immunocompetent mice. This system additionally provided the opportunity to investigate whether CpG or UpA elevation showed comparable phenotypic effects on a virus with a substantially different replication mechanism to echovirus 7 (E7), which is markedly attenuated upon increase of the CpG or UpA dinucleotide frequency of its genome (*Atkinson et al., 2014*).

## Results

### Generation of IAV mutants with maximised CpG and UpA dinucleotide composition

Segment 5 was selected for mutagenesis as it showed no evidence for alternative reading frames, manifested by an absence of suppressed synonymous site variability (*Figure 1—figure supplement 1A*) apart from the extreme 3'end of the gene, likely reflecting the presence of a segment-specific RNA packaging signal (*Gog et al., 2007*; *Hutchinson et al., 2010*).

Segment 5 showed marked suppression of CpG and UpA frequencies across the segment (*Figure 1—figure supplement 1B*), with averaged values over a window size of 120 bases consistently below the expected frequency of 0.054 based on mononucleotide composition (freq. G * freq. C). The mean observed CpG frequency of 0.026 collectively for viruses infecting human swine or avian hosts was 46% of the expected value (O/E ratio). UpA frequencies were similarly suppressed, with a mean observed frequency of 0.031 compared to an expected value of 0.068 (O/E ratio: 0.46).

The program Sequence Mutate in the SSE package (*Simmonds, 2012*) was used to modify CpG and UpA frequencies in segment 5 without altering protein coding whilst using a recently developed option to maintain mononucleotide composition through the introduction of compensatory substitutions elsewhere in the sequence. Increases in UpA frequencies in sequences maximised for CpG content were re-normalised for UpA composition, and *vice versa*. Mutagenesis was carried out between positions 151–1413 of segment 5 to avoid any possible disruption of packaging signals or replication/translation elements at the terminal 150 bases at the ends of the segment (*Gog et al., 2007*; *Hutchinson et al., 2010*; *Ozawa et al., 2007*).

The resulting CpG-high mutant contained 86 additional CpG dinucleotides, but only one extra UpA and an identical G+C and mononucleotide content to WT virus (*Table 1*). The UpA-high mutant contained an additional 73 UpA dinucleotides, and was similarly normalised for CpG and mononucleotide composition. A permuted mutant (CDLR) with identical coding and dinucleotide frequencies to WT virus was generated as described previously (*Atkinson et al., 2014*). Mutants showed minimal changes (CDLR, CpG-high) or modest reductions (UpA-high) in codon pair scores.

### Replication phenotypes of segment 5 CpG- and UpA-high mutants

Virus stocks with WT, CDLR, CpG-high and UpA-high segment 5 sequences were produced by transfection of 293T cells with pDUAL plasmids and amplified and titrated for infectivity in MDCK cells and virions quantified by haemagglutination (HA) assay. CpG-high and UpA-high mutants showed

**Table 1.** Composition and coding parameters of the mutated region of segment 5.

| | Subs.[a] | C+G% | CpG | ΔCpG[b] | CpG-O/E[c] | UpA | ΔUpA[b] | UpA-O/E | CAI | CPS[d] |
|---|---|---|---|---|---|---|---|---|---|---|
| PR8 WT | — | 0.46 | 28 | — | 0.43 | 43 | — | 0.49 | 0.745 | 0.005 |
| CDLR | 134 | 0.46 | 28 | 0 | 0.43 | 43 | 0 | 0.49 | 0.745 | 0.011 |
| CpG-high | 233 | 0.46 | 114 | +86 | 1.63 | 45 | +2 | 0.51 | 0.611 | -0.011 |
| UpA-high | 199 | 0.46 | 29 | +1 | 0.56 | 116 | +73 | 1.31 | 0.627 | -0.118 |

[a] Number of sequence changes from WT sequence
[b] Change in the numbers of CpG and UpA dinucleotides
[c] Observed to expected frequencies of CpG and UpA dinucleotides
[d] Calculated as previously described (**Buchan et al., 2006**)

slower replication kinetics in multistep replication assays in MDCK cells (*Figure 1A*), with approximately ten-fold less infectious virus produced at the 24 hr time point. Titering of multiple replicates (n = 5) at 24 hr p.i. in replicate experiments demonstrated significant differences in titre between WT and both CpG- and UpA-high mutants (p = 0.009 and p = 0.014 respectively; *Figure 1B*), while replication of WT and CDLR was equivalent ($p \approx 0.5$). Similar magnitude replication deficits of the CpG and UpA mutants were also seen in single cycle infections in MDCK cells (*Figure 1—figure supplement 2*)

Additional evidence for an impaired replication ability of these mutants was obtained by the observation of consistently smaller plaque sizes of CpG-high (0.061 ± 0.006 cm) and UpA-high (0.039 ± 0.003 cm) mutants than WT or permuted virus (0.144 ± 0.011 cm and 0.132 ± 0.010 cm respectively; *Figure 1C*). Virions of CpG- and UpA-high IV mutants showed reduced infectivity compared to WT (and CDLR) variants, with approximately three to four-fold elevated haemagglutinin (HA) to infectivity ratios in virus stocks grown in embryonated eggs (*Figure 1D*). Comparable differences between WT and mutant forms of IAV were observed for RNA/infectivity ratios using a quantitative PCR (qPCR) for segment 5 sequences (*Figure 1E*).

To investigate whether differences in virion/infectivity ratios in IAV variants with compositionally altered segment 5 sequences were the result of packaging defect on the mutant segment, we infected MDCK cells at low MOI with WT, CDLR, CpG- and UpA-high mutants of IAV and compared frequencies of cells expressing individual proteins by immunocytochemistry at 6 hr post infection (*Figure 1—figure supplement 3*). The relative frequency of WT-infected cells expressing NP (encoded by segment 5) to those expressing viral proteins M2, NS1, NA and PB2 were comparable to those of CDLR, CpG-high and UpA-high mutants. Similarly, polyacrylaminde gel electrophoresis (PAGE) of purified egg-derived virions of WT and mutant strains revealed similar relative proportions of IAV structural proteins (*Figure 1—figure supplement 4*). Finally, the relative amounts of segment 5 and segment 2 RNA was compared in purified virions was quantified by qPCR (*Figure 1—figure supplement 5*). RNA ratios in WT were comparable in other IAV variants. Combined, these three methods of analysis provided no evidence for a packaging defect of segment 5 in CpG- and UpA-high virions and do not explain the reduced infectivity and replication kinetics of CpG- and UpA-variants of IAV.

Replication differences between WT and mutant IAV viruses were closely reproduced in virus competition assays, in which equal infectivities of WT virus and mutant viruses were co-cultured over 5–10 high multiplicity passages (*Figure 1—figure supplement 6*). Cleavage of sequences amplified from segment 5 with restriction enzymes that differentiated between mutants (*Table 3*) was followed by gel electrophoresis and densitometry to quantify viral populations. WT and permuted viruses fully outcompeted CpG-high and UpA-high mutants by passage 5 while UpA-high and CpG-high mutants were equally fit (equimolar by passage 10) while WT showed marginally greater fitness than CDLR (WT: 60%; CDLR: 40% at passage 10). The overall fitness ranking inferred from competition assays was PR8-WT ≥ >> UpA-high = CpG-high.

Overall, the in vitro findings revealed that elevation of either CpG or UpA frequencies in one segment of IAV substantially reduced replication fitness. The ten fold reductions in replication titres in the growth curve in mutants with approximately 10% genome replacement (*Figure 1A, B*) were broadly consistent with fitness reductions associated with CpG and UpA frequency increases in E7

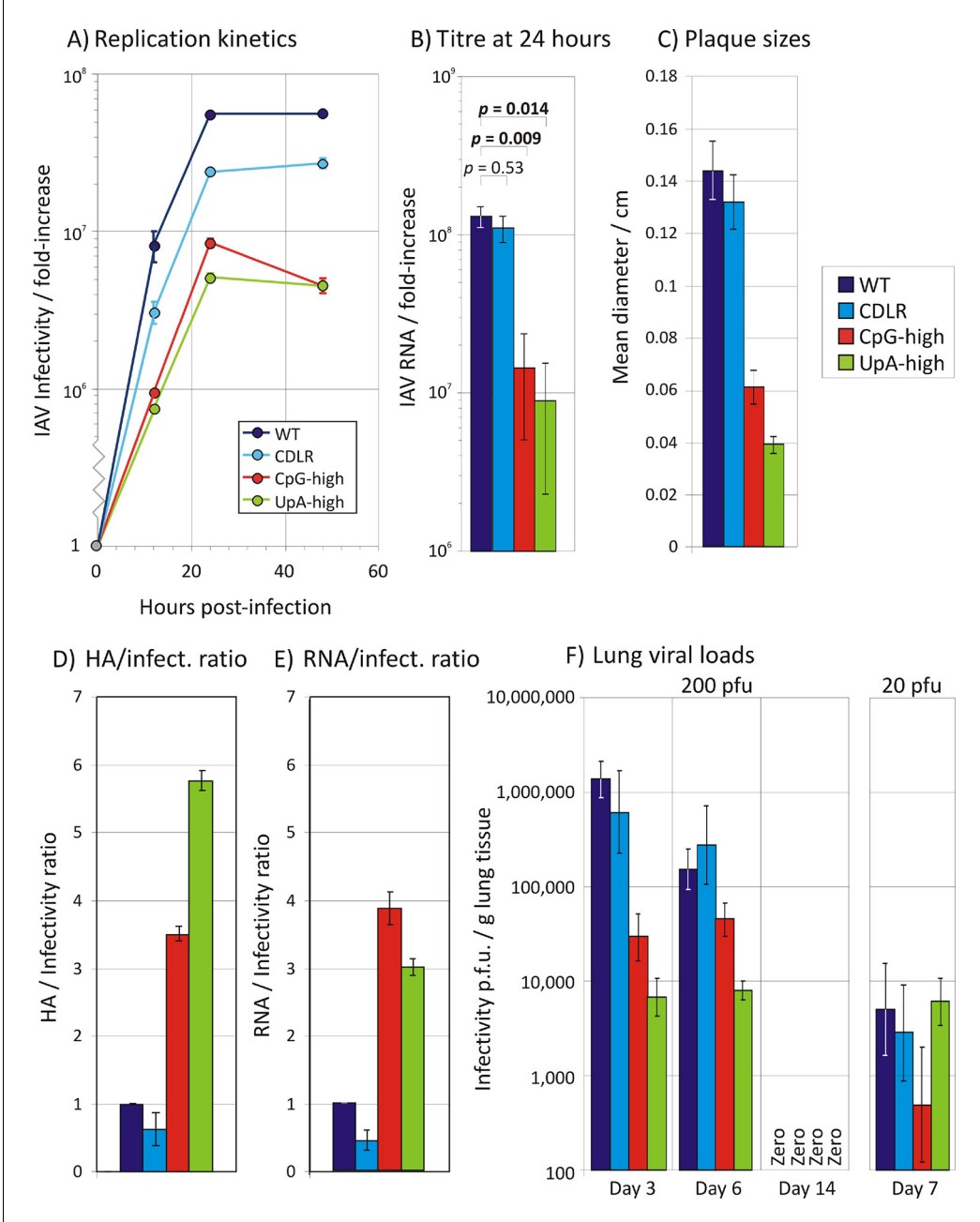

**Figure 1.** Replication phenotypes of IAV WT and compositionally altered mutants. (**A**) Replication kinetics of IAV in a multi-cycle replication assay; MDCK cells were infected at an MOI of 0.01 and supernatant assayed for IAV RNA at three post-infection time points; y-axis records infectivity of supernatant on titration in MDCK cells. Error bars show SEM of 3 replicate assays. A single replication cycle assay is shown in *Figure 1—figure supplement 1*. (**B**) Infectivity titres of supernatants collected at 24 hr. Bars show geometric titres of 5 replicate cultures of WT, CDLR, CpG-high and UpA-high variants; error bars show SEMs. The significance of titres differences was determined by Kruskall-Wallis non-parametric test; *p* values shown above bars with significant values highlighted in bold. (**C**) Mean plaque diameters of approximately 30 plaques of WT and mutant IAV variants; error bars show SEMs. (**D, E**) HA and RNA/infectivity ratios of WT and permuted, CpG-high and UpA-high mutants in MDCK cells; bar heights show mean values from 2–3 replicate assays; error bars show standard errors of the mean (SEM). (**F**) Infectivity titres of lung homogenates collected at days 3, 6 and 14 (experiment 2; *Figure 2*) and day 7 from mice infected with 20 PFU (experiment 3) from inoculated mice determined by titration on MDCK cells. Bar heights show mean values from cohorts of 4–6 mice; error bars show SEM. Synonymous site variability and composition analysis of IAV segment 5 is shown in *Figure 1—figure supplement 1*. Replication kinetics of IAV in a single cycle / high MOI

*Figure 1 continued on next page*

*Figure 1 continued*

replication assay is shown in *Figure 1—figure supplement 2*. Expression of IAV viral proteins M2, NS1, NA and PB2 relative to that of NP in different IAV mutants at 6 hr post-infection is shown in *Figure 1—figure supplement 3*. Detection of IAV viral proteins HA1, M1 and HA2 relative to that of NP in purified virions from different IAV mutants is shown in *Figure 1—figure supplement 4*. Ratio of segment 5 and segment 2 RNA sequences in purified virions is shown in *Figure 1—figure supplement 5*. Pairwise comparisons of the replication fitness of the mutants by competition assays is shown in *Figure 1—figure supplement 6*.

The following figure supplements are available for figure 1:

**Figure supplement 1.** Variability and composition analysis of IAV segment 5.

**Figure supplement 2.** Single replication cycle kinetics of IAV infected an am MOI of 5 Replication kinetics of IAV in a single-cycle replication assay; MDCK cells were infected at an MOI of 5 and supernatant assayed for IAV RNA at 5 post-infection time points; y-axis records infectivity of supernatant on titration in MDCK cells.

**Figure supplement 3.** Expression of IAV viral proteins M2, NS1, NA and PB2 relative to that of NP in different IAV mutants.

**Figure supplement 4.** Detection of IAV viral proteins HA1, M1 and HA2 relative to that of NP in purified virions of different IAV mutants.

**Figure supplement 5.** Ratio of segment 5 and segment 2 RNA sequences in purified virions Quantitation of segment 5 and segment 2 RNA by qPCR in purified virions of WT and mutant IAV strains.

**Figure supplement 6.** Pairwise comparisons of the replication fitness of the mutants by competition assays.

---

(*Atkinson et al., 2014*) and (via codon pair de-optimisation) in IAV (*Mueller et al., 2010*) and poliovirus (*Coleman et al., 2008*).

## In vivo clinical course of IAV mutants with increased CpG and UpA frequencies in vivo

Induction of disease in mice infected with WT IAV and variants with altered dinucleotide frequencies was investigated by infection of immunocompetent 8 week old female BALB/c mice. Intranasal inoculation of groups of 6 mice with 200 PFU of WT PR8 and CDLR permuted mutant induced rapid weight loss (down to 86.3% of starting weight, standard error ± 3.2% and 86.5% ± 1.5%) by day 5 (*Figure 2A*). Mice showed increasing clinical signs including reduced activity, loss of condition (staring coat), increased respiration and hunching over the observation period. Mouse cohorts infected with UpA-high and CpG-high mutants showed reduced weight loss (92.3% ± 2.0% and 96.4% ± 2.7% of starting weight respectively) and reduced (UpA-high) or absent (CpG-high) clinical signs compared to WT-infected mice.

Although WT-infected and CDLR infected mice showed rapid weight loss, those infected with UpA- and particularly CpG-high IAV mutants showed a less severe clinical course. To investigate potential differences in recovery rates between WT and mutants and to examine virus replication and host response at different time points, three cohorts of 6 mice were inoculated each with WT, CDLR, CpG- and UpA-high variants of IAV and culled at days 3, 6 and 14 (Experiment 2; *Figure 2B*). Initial weight loss and clinical severity over the first 5 days was reproducible from the previous experiment with again CpG-high infected mice both strikingly reduced weight loss compared to WT and CDLR controls. Over the period after 5 days, individual mice from the WT, CDLR and UpA-high which showed >20% weight loss and more severe clinical signs and were culled. The remainder, along with all CpG-high infected mice regained weight from a nadir at around day 7–8; the latter group regained weight similar to those of the mock infected group while remaining mice retained a weight deficit through to day 14. In contrast to WT, CDLR and UpA-high infected mice, all mice infected with the CpG-high mutant remained clinically normal throughout the infection period and showed minimal weight loss over the infection period (maximum loss to 95.8% of starting weight, SE ± 1.5%).

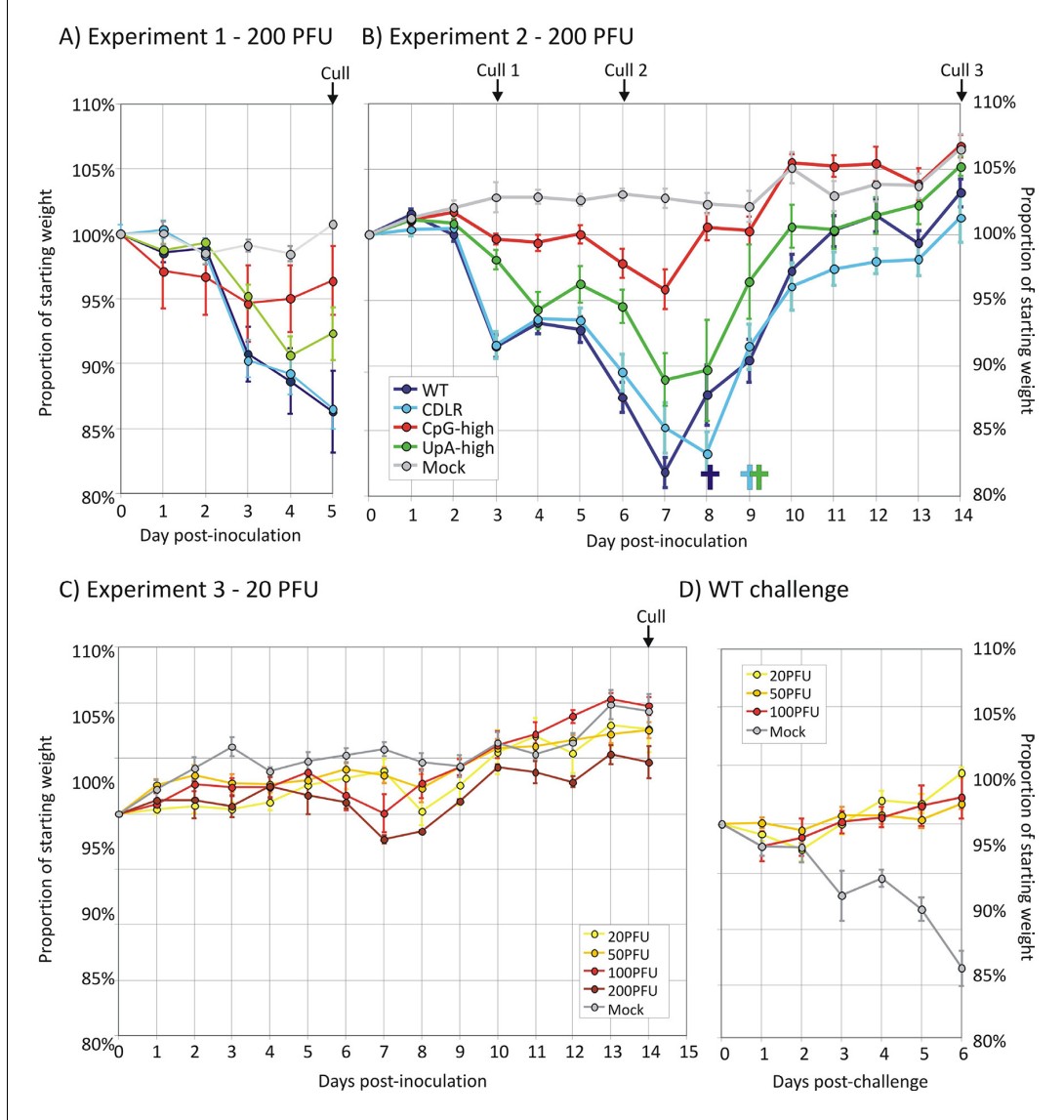

**Figure 2.** Infection outcomes and protective immunity in mice infected with IAV. (**A**, **B**) Weights of mice (proportion of starting weight) of mice inoculated with 200 PFU of IAV WT and mutant strains with altered dinucleotide compositions. Deaths of individual mice are shown underneath the x-axis using the same colour coding. (**C**) Weight loss of mice infected with different inoculum doses (20–200 PFU) of the CpG-high IAV mutant (**D**) weight loss in the mice previously infected with CpG-high IAV (in *Figure 2C*) with a 200 PFU WT challenge dose at day 21 after the original inoculation. In all graphs, error bars show SEMs of 4–6 mice per group.

## IAV virus replication in vivo

To investigate the extent of IAV replication in the respiratory tract of inoculated mice, tissue homogenates were prepared from individual lungs of mice culled at days 3, 6 and 14 and titrated for infectivity in MDCK cells and for viral RNA by qPCR (*Figure 1F*). WT and the CDLR mutant showed similar replication kinetics while there was marked replication deficit of CpG-high and UpA-high mutants, most evident at day 3 but remaining over 1 log below WT levels at day 6. For all mouse groups (WT, CDLR, UpA-high, CpG-high), IAV replication was undetectable by day 14, with all homogenates testing negative by virus isolation (*Figure 1F*) and by qPCR (data not shown).

## Histopathology and innate host response to IAV infection

The histopathological changes associated with IAV infection in the lung were examined at three time points (days 3, 6 and 14 post-inoculation) in mock infected mice and mice inoculated with 200pfu of WT, CDLR, CpG- and UpA-high variants of IAV. Histological examination revealed mild to marked, multifocal to coalescing, fibrinonecrotising bronchointerstitial pneumonia typical of IAV infection (*Figure 3A*). Changes were consistent with an acute to subacute disease process at days 3 and 6 while day 14 samples exhibited more chronic pathology including prominent evidence of repair such as piling up of epithelial cells lining the airways and florid type II pneumocyte proliferation outlining alveolar spaces. Infection with the CpG-high mutant virus produced less pathology at day 3 compared to the WT, CDLR and UpA-high variants. The CpG-high mutant also showed a significantly more rapid resolution of inflammation (*Figure 3B*; scored as the mean of perivascular, peribronchiolar and interstitial inflammatory lesions; p = 0.01). Epithelial and interstitial necrosis was prominent in mice infected with all IAV variants, but had largely resolved by day 14 (*Figure 3C*). Consistent with its reduced overall pathology, CpG-high showed more rapid resolution of epithelial and interstitial cell repair processes at day 14 (*Figure 3D*; p = 0.01). Contrastingly, the histopathological changes in the UpA-high infected mice were similar to WT and CDLR infected mice at all time points and dosages.

The induction of the major cytokines during acute infection with IAV was determined in pooled samples of lung homogenate from cohort groups at days 3, 6 and 14 after inoculation (*Figure 3—figure supplement 1*). Substantial (>two-fold) induction of 21 of the 40 cytokines in the panel over levels in mock infected mice was observed in at least one IAV strain at days 3 and 6, but with these showed substantial variability in levels between time points. With the exception of IP-10, which was maximally induced at day 3, cytokine levels increased between days 3 and 6, while they became largely undetectable on day 14 after resolution of infection. Most prominent among induced cytokines were chemokines (*e.g.* RANTES, MIP-1α) and less prominent induction of interleukins or TNF-α. In contrast to the increased levels between days 3 and 6 in the majority of the inflammatory cytokines, individual mouse measurements of IFN-β mRNA by qPCR showed consistently higher levels at day 3, particularly in the WT and CDLR-infected groups (*Figure 3—figure supplement 2*). Induction of cytokines was comparable in Experiment 1 on day 5 post-inoculation (*Figure 3—figure supplement 3*). Differences were observed between WT and CDLR groups compared with the CpG-high and UpA-high groups. WT (and CDLR) variants of IAV showed greater induction of cytokines on day 3 (BLC, I-TAC, JE, MIG, MIP-1α and 1β) while by day 6, responses were comparable to those of the clinically attenuated CpG-high- and UpA-high mutants. MIP-1α was induced to substantially higher levels in mice infected with the latter viruses on day 6.

## Adaptive immune response to IAV infection

T cell responses to IAV infection at day 21 post-inoculation were quantified using ICS and flow cytometry by measuring frequencies of ex vivo CD8+ and CD4+ T lymphocytes producing the cytokines IFN-γ, TNF-alpha or IL-2 upon exposure to pooled immunoreactive peptides in the NP (n = 1) and HA (n = 3) proteins. Lymphocytes extracted from lung (representing the local response) and spleen (systemic response) at day 21 post-infection were gated using the strategy outlined in *Figure 4—figure supplement 1*.

Infection of mice with IAV induced a strong IFN-γ response in the lungs of infected mice, with frequencies of antigen-specific CD8+ lymphocytes of at least 5% in WT and CDLR-inoculated mice (*Figure 4A, B*). The population of IFN-γ producing CD8+ T cells in CpG-high mice was comparable (3.5%), while UpA infection induced a lower frequency of IFN-γ producing CD8+ T cells (1.2%). All variants of IAV induced comparable IFN-γ responses to IAV in CD8+ T lymphocytes recovered from the spleen with a 200 PFU inoculum (1.0–1.4%; *Figure 4B*). These lymphocytes also showed induction of the cytokine TNF-α and IL-2 that was comparable between WT and CDLR, CpG- and UpA-high IAV mutants.

Likewise peptide-specific CD4+ T cell responses were generated in all groups infected with variant or wild-type IAVs, directed at the single HLA Class II-restricted epitope included in the peptide cocktail. Overall the CD4+ T lymphocyte peptide-specific response recorded both locally (lung) and systemically (spleen) was substantially weaker (< 0.5%) than that of CD8+ T cells (data not shown). In order to determine if infection by the attenuated viruses had effects on the hierarchy or magnitude

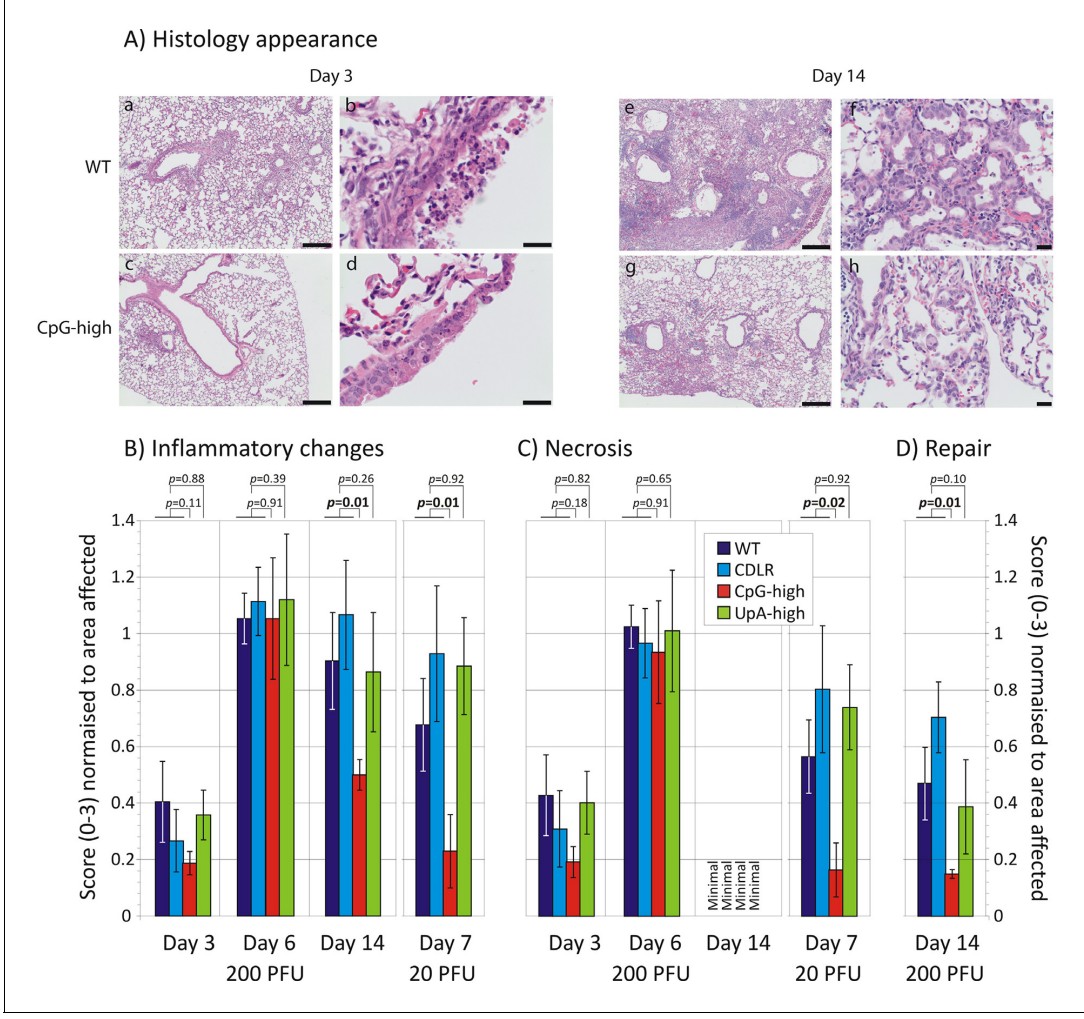

**Figure 3.** Cytopathology and innate immune responses to IAV infection in mice. (**A**) Representative lung sections from mice infected with WT (a, b, e, f) and CpG-high (c, d, g, h) IAV variants at days 3 (a-d) and 14 (e-h) post infection. At day 3 prominent peribronchiolar and perivascular accumulation of inflammatory cells are present in WT infected mice (a), with moderate to marked, multifocal to coalescing airway epithelial cell necrosis (b). The inflammatory and necrotic lesions in this acute phase of disease are less severe in the CpG-high infected mice (c, d). At day 14 WT-infected mice showed marked peribronchial and perivascular lymphohistiocytic inflammation (e), epithelial regeneration and prominent type II pneumocyte proliferation (f). Again, the lesions in the CpG-high infected mice during this repair stage of disease were less severe (g, h). Bars in figures a, c, e, and g represent 200 μm. Bars in figure b, d, f, and h represent 20 μm. (**B**, **C** and **D**) Blinded histological scoring of (**B**) inflammation, (**C**) necrosis, and (**D**) repair processes in sections of lung from mice infected with different IAV strains. Bar heights show mean values from 4–6 mice per group scored from 0–3 for severity of (**B**) inflammation (perivascular, peribronchiolar and interstitial), (**C**) epithelial and interstitial necrosis, and (**D**) epithelial cell repair and type II pneumocyte proliferation. All average scores were normalised by the area of lung affected in the section. The significance of differences in pathology severity between viruses was determined by Kruskall-Wallis non-parametric test (combining WT and CDLR scores); *p* values shown above bars with significant values highlighted in bold. Error bars show SEM. Induction of individual cytokine in mice infected with IAV infection at different time point post-inoculation is shown in *Figure 3—figure supplement 1* and *3*. Induction of interferon-β mRNA in lung samples is shown in *Figure 3—figure supplement 2*.

The following figure supplements are available for figure 3:

**Figure supplement 1.** Cytokine response to IAV infection at different time point post-inoculation.

**Figure supplement 2.** Induction of interferon-β mRNA in lung samples.

**Figure supplement 3.** Cytokine responses in lungs of mice infected in experiment 1 (day 5).

of the T cell response to individual epitopes, ELISPOT assays were performed on thawed frozen

splenocytes from mice infected with 200 PFU to quantify their responses against individual peptides used in the ICS cocktail (*Figure 4—figure supplement 2*). Despite alterations to the NP coding sequence, the CpG-high, UpA-high and CDLR groups generated memory CD8+ T cell responses to the dominant NP epitope of equivalent magnitude to the wild-type infected animals. CD8+ and CD4+ cell responses to the unaltered HA protein were unchanged and infection by the attenuated viruses did not appear to alter the hierarchy of dominant epitopes in the peptides measured.

Induction of neutralising antibody to the PR8 strain of IAV was measured at days 6 and 21 post inoculation (*Figure 4B*). High titres of antibody were induced at day 21 with comparable end point titres observed between WT/CDLR and the attenuated CpG and UpA-high IAV mutants at both measured time points.

## Immunoreactivity of CpG- and UpA-high mutants

To investigate the extent to which infection dose influenced host response, cohorts of 6 mice were infected with 20 PFU of WT, CDLR, CpG-high and UpA-high IAV strains. Mice showed minimal weight loss and clinical signs over the infection period (data not shown) and consistent ten- to hundred-fold reduction in lung viral loads as determined by infectivity and quantitative PCR measurements (*Figure 1F*). This was associated with reduced pathology changes in the lung, with consistent reductions in mean scores for epithelial damage and inflammatory cell infiltrates across all variants of IAV (*Figure 3B, C*). Significant differences were observed in the severity of pathology between CpG-high and control mice (WT and CDLR groups). The percentage of peptide specific CD8+ and CD4+ T cell responses in the mice given ten-fold lower doses (20 PFU) was correspondingly reduced in all the groups, with an approximately five-fold reduction in local (lung) memory responses and a three-fold reduction in the central (splenic) memory compartment as measured by IFN-γ production (*Figure 4B, C*). Nonetheless, it is notable that even at the low dose all the modified IAV strains were able to generate T cell memory responses of comparable magnitude to that observed in the wild-type IAV-infected mice. Induction of neutralising antibody, in contrast, was only minimally reduced in mice infected with 20 PFU compared to the original inoculating dose (*Figure 4B, C*).

Overall, with the exception of neutralising antibody induction, there was a strong dose dependence on all measured metrics of outcomes and host responses to IAV infection. These findings contrast markedly with the maintained immunoreactivity of CpG- and UpA-high mutants of IAV; despite their impaired replication in vivo (*Figure 1F*), they maintained comparable innate and T cell responses to infection.

## Use of attenuated IAV variants as vaccines

Mice infected with the original 200 PFU dose of the CpG-high mutant showed an almost entirely non-pathogenic course of infection (absent clinical signs, 4% weight loss), ten–hundred fold lower viral loads (*Figure 1F*) and reduced lung pathology (*Figure 3*) and yet induced substantial host responses, similar to those of WT virus. This raised the question of whether the response to infection with the CpG high mutant conferred protection from re-infection. To investigate this and to determine the dose of virus required to confer protection, cohorts of 4 mice were infected with reduced doses (100, 50 and 20 PFU) of the CpG-high virus mutant and weight loss was compared with those infected at the original dose (Experiment 3 - *Figure 2C*). Infection with the 200 PFU dose reproduced the minimal weight loss observed previously at day 6 (97.7% ± 0.4% of starting weight compared to 96.4% ± 2.7% previously observed [*Figure 2A, B*]), while mice infected with all lower doses showed no significant weight loss relative to the uninfected control group. No mice showed clinical signs of infection.

At day 21 post-inoculation, mice in each group (200 PFU, 100 PFU, 50 PFU, 20 PFU and mock) were challenged with 200 PFU of PR8 WT virus. As expected, the mock-immunised group developed severe symptoms of infection comparable to the outcome of WT infection observed in previous experiments (*Figure 3D*). In contrast, none of the mice previously inoculated with any dose of CpG-high virus showed clinical symptoms of infection or weight loss. Lung samples collected at day 6 post-WT inoculation with all doses of CpG-high IAV were consistently virus and PCR-negative, indicating that sterilising immunity had been achieved.

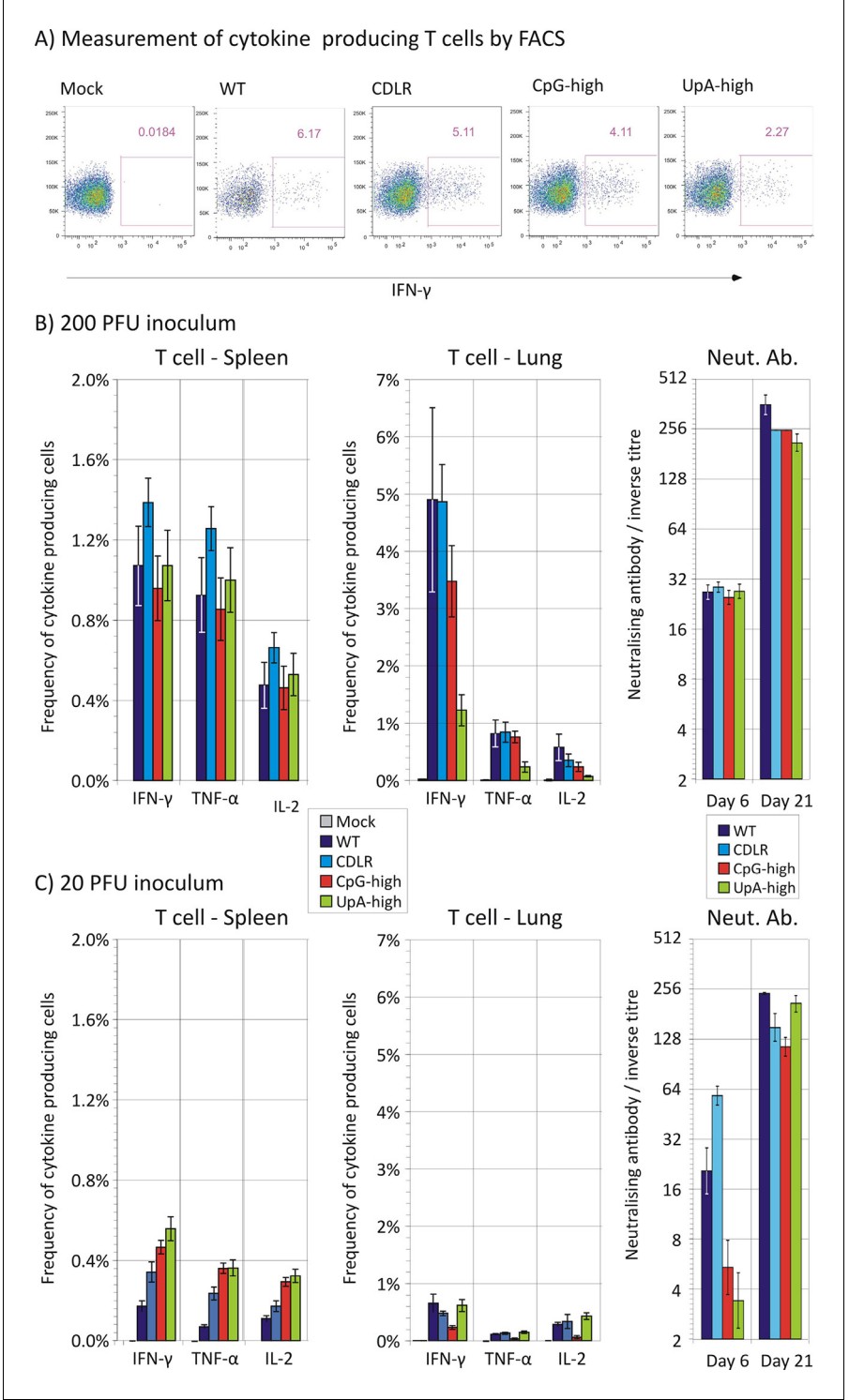

**Figure 4.** Adaptive immune response (T cell and neutralising antibody) after infection with wild-type and mutant strains of IAV. (**A**) Representative FACS plots showing the percentage of IFN-γ producing cells in mice infected with 200 PFU of IAV in the CD8+ T cell population after peptide stimulation. (**B, C**) Mean frequencies of CD8+ T lymphocytes expressing IFN-γ, TNF-α or IL-2 from spleen and lung at day 21 and neutralising antibody titres at days 6 and 21 post-infection with 200 PFU or 20 PFU of IAV respectively. Bars show mean frequencies of lymphocyte subsets and antibody titres from 4–6 mice per group; error bars show SEM. The gating strategy to identify CD8 lymphocytes is shown in *Figure 4—figure supplement 1*. Reactivity to individual peptides is shown in *Figure 4—figure supplement 2*.

*Figure 4 continued on next page*

*Figure 4 continued*

The following figure supplements are available for figure 4:

**Figure supplement 1.** Gating strategy to identify and quantify lymphocytes producing IFN-γ.

**Figure supplement 2.** ELISPOT analysis of T cell reactivity to individual IAV peptides.

## Discussion

Increasing the frequencies of CpG and UpA dinucleotides in segment 5 of the PR8 strain transformed the phenotype of IAV in cell culture and in vivo. CpG- and UpA-high variants were attenuated in vitro and showed major alterations in pathogenicity on mouse inoculation. The approximately ten-fold reduction in replication kinetics reproduced the attenuation achieved in different classes of viruses where similar degrees of mutagenesis had been applied (see *Table 1* in *Tulloch et al., 2014*), suggesting that attenuation originates through a shared, although as yet immunologically uncharacterised mechanism. Quantitation of virion protein expression and production and of RNA copies from different IAV segments in viral stocks of WT and CDLR, CpG- and UpA-high mutants provided evidence that the mutated segment 5 viral RNA and protein were packaged appropriately (*Figure 1—figure supplements 3*, *4* and *5*).

The similarity in replication kinetics between WT and the permuted control, CDLR furthermore indicated that the attenuated phenotype of the CpG- and UpA-high mutants originated from compositional alterations in the coding sequence and was not the result of impaired translation (*Mueller et al., 2010*; *Coleman et al., 2008*; *Martrus et al., 2013*; *Ni et al., 2014*; *Shen et al., 2015*; *Simmonds et al., 2015*), unintended disruption of uncharacterised RNA structure or replication elements embedded in the segment 5 coding sequence. Segment 5 is indeed not suspected of coding for additional proteins in alternative reading frames and shows little evidence of RNA structure or other elements in the central coding region that are required for replication (*Gog et al., 2007*; *Hutchinson et al., 2010*; *Moss et al., 2011*; *Priore et al., 2012*; *Figure 1—figure supplement 1*). The mutagenesis applied left codon pair bias relatively unchanged; for example the CPS of the CpG-High mutant was -0.011 compared to a WT score of 0.011 (*Table 1*). Use of the permuted control, maintenance of the CPS score in mutants, verification of unimpaired transcriptional and translational profiles in virus stocks and application of mutagenesis in a region without alternative transcript production provides a sufficiently robust framework for us to be confident that the observed attenuation of the CpG- and UpA-high viruses was primarily the result of the introduction of CpG and UpA dinucleotides.

### Development of an attenuated vaccine for influenza A virus

The CpG-high virus was replication defective and infections in mice were associated with a clear reduction in pathogenicity compared to WT and oterIAV mutants. However, the CpG-high mutants induced comparable immune and cytokine responses to WT virus, including potent T cell responses both in the lung and spleen (*Figure 4B*), equivalent to the response to WT virus despite its >ten-fold reduction in replication. In contrast, the UpA-high virus also showed restricted replication yet infections in mice were accompanied by substantial pathogenicity, including weight loss, clinical signs of infection and inflammatory/necrotic damage to the lung comparable to WT virus.

Several attributes of the CpG-high mutant produced in the current study are therefore favourable for the potential future use of this attenuation strategy to produce an effective and safe vaccine for IAV. Firstly, we have documented that its attenuated phenotype results directly from the compositional changes in segment 5 since other forms of sequence disruption that retained dinucleotide frequencies (CDLR) was phenotypically equivalent to WT. Thus attenuation is based on the cumulative effect of several hundred individual nucleotide changes associated with increasing CpG and UpA dinucleotide frequencies that cannot readily revert to wild type virulence. This contrasts with conventional attenuation mechanisms, in which replication ability is decreased through the presence of a limited number of nucleotide changes that may potentially revert. For example, attenuation of the currently used temperature sensitive FluMist backbone of IAV is dependent on only 4–5 amino acid

changes (*Jin et al., 2003*) and there is at least the theoretical possibility that one or more of these may revert and increase virulence in vaccinees.

As reviewed previously (*Simonsen et al., 2005*; *Rimmelzwaan et al., 2007*; *Sridhar et al., 2013*)., attenuated vaccines share with inactivated IAV vaccines an ability to induce powerful T cell responses that represent an important element of the protective immunity to wild type challenge. The CpG-mutant in the current study induced these responses both in the lung and spleen (*Figure 4B*), equivalent to the response to WT virus despite its >ten-fold reduction in replication in the lung. The presence of T cell epitopes in more conserved genome regions than HA and NA additionally provides the ability of a monovalent attenuated vaccine to confer protection against heterologous IAV challenge, as has been demonstrated in experimental mouse models (*Nogales et al., 2014*; *Wang et al., 2009*).

Finally, while both CpG- and UpA-high mutants showed replication impairment in cell culture and in vivo, the CpG-high mutant showed substantially greater attenuation of pathogenicity, despite the greater restriction of replication of UpA-high in vivo These findings suggest the existence of additional host response factors and interactions beyond replication phenotype in determining outcomes of infection. These may potentially be related to the ability of both CpG- and UpA-high mutants to induce powerful innate and adaptive immune responses in infected mice disproportionate to their replication phenotypes. The key observation was the equivalence in T cell and B cell responses and induction of local cytokine responses by virus mutants that replicated ten–hundred fold less in the lung than WT (or CDLR) viruses. Contrastingly, inoculation of all of these viruses at a ten-fold lower dose reduced host response to a degree comparable to their reduction in replication. Similar maintenance of host response despite attenuation of replication has been observed repeatedly in other studies in which codon or codon pair usage was altered (*Yang et al., 2013*; *Nogales et al., 2014*; *Mueller et al., 2010*). As discussed above, these parallel outcomes would be anticipated if their replication impairment arose though unintended increases in CpG and UpA frequencies. Contrastingly, observations of their high levels of apparently enhanced immunogenicity would furthermore be highly problematic to explain had their attenuation been dependent purely on inhibition of replication through retardation of translation.

Understanding the pathways by which CpG- and UpA-high virus mutants are attenuated is essential before dinucleotide compositional alteration can be adopted as generic vaccine strategy. As discussed previously (*Simmonds et al., 2015*), attenuation is dependent more on an active restriction pathway within the cell rather than CpG- and UpA-high mutants being intrinsically defective. In the future development of CpG-high mutants as attenuated vaccines, further work is clearly required to understand how this compositional change can uncouple the three outcomes of replication kinetics, host immune response and pathogenicity. The apparent "adjuvanting" effect of increased CpG and UpA dinucleotides on host immune response observed in the current study is highly relevant to the balance between vaccine efficacy and safety. Documenting the pathway(s) by which the cellular restriction of replication observed in cell culture (*Burns et al., 2009*; *Atkinson et al., 2014*; *Burns et al., 2006*) mediates the enhanced inflammatory and adaptive immune responses observed in vivo is important in understanding this phenomenon.

## Materials and methods

### Cell culture and cell lines

Human embryonic kidney (293T) cells, human alveolar basal epithelial (A549) cells and Madin-Darby canine kidney (MDCK) cells were cultured in Dulbecco's modified Eagle's medium (DMEM) supplemented with 10% foetal bovine serum (FBS), penicillin (100 U/ml) and streptomycin (100 µg/ml). All cells were maintained at 37°C with 5% $CO_2$. All three cells were tested at 3–6 month intervals for mycoplasma contamination by a commercial PCR-based method and found negative. The identity of A549 and MDCK cells was verified by their sialic acid repertoires and at the species level by their differential sensitivity to human β-interferon. Cell lines were originally provided by the National Institute for Medical Research, Mill Hill, London, UK.

## In silico construction of mutants with modified segment 5 sequences

The MDCK-adapted PR8 laboratory strain cloned by de Wit and colleagues (*de Wit et al., 2004*) was used as the reference genome in the construction of all mutants. CpG and UpA frequencies were separately maximised between positions 151 and 1514 in segment 5 (NP) of the PR8 strain (Genbank accession number EF467822) while preserving coding using the Program Sequence Mutate in the SSE package (*Simmonds, 2012*) (Table S1). As a control, a permuted mutant was designed using the program CDLR in the SSE package that scrambles the order of codons encoding the same amino acids but preserves dinucleotide frequencies, codon usage and encoded amino acid sequence. DNA constructs of the designed sequences were synthesised (GeneArt; sequences listed in the *Supplementary file 1*). For all sequences, ratios of observed CpG and UpA frequencies to expected frequencies derived from mononucleotide composition were calculated by the formula $f(XpY)/f(X)*f(Y)$ and referred to as O/E ratios (*Table 1*). The codon adaptation index for human codon usage was calculated through the website http://genomes.urv.es/CAIcal/ (*Puigbo et al., 2008*).

## Recovery of mutant IAVs

IAV particles were generated using reverse genetic approach as described previously (*Hutchinson et al., 2008*). Firstly, sequences of the mutated segment 5 sequences were assembled into separate bidirectional expression ("pDUAL") plasmids. Then, 8 pDUAL plasmids (250 ng each with 4 µl Lipofectamine 2000 (Life Technologies, Loughborough, UK) containing sequences of a complete IAV genome were used to transfect 293T cells simultaneously, with transfections either incorporating the WT segment 5 sequence or a dinucleotide modified version. After overnight incubation, virus growth medium (DMEM supplemented with 5 µg/ml TPCK-treated trypsin, 0.14% BSA fraction V and penicillin (100 U/ml) and streptomycin (100 µg/ml)) was added to allow a small-scale amplification of the viruses in 293T cells. After 48 hr, the virus particle-containing supernatants were passaged on MDCK cells or in embryonated chicken eggs to further amplify the viruses to obtain working stocks. To confirm the sequences of recovered IAV mutants were correct, RNA was extracted from the virus-containing supernatants using QIAamp Viral RNA Mini Kits (Qiagen, Manchester, UK). A reverse transcription reaction was performed using SuperScript II Reverse Transcriptase (Life Technologies, Loughborough, UK). Subsequently, the cDNA was amplified by polymerase chain reaction (PCR) using segment 5-specific primes listed in *Table 2*, and sequenced by BigDye Terminator v3.1 Cycle Sequencing Kit (Life Technologies, Loughborough, UK).

## Infectivity titrations

Virus titres were determined by plaque titration using ten-fold serial dilutions of virus stocks. Confluent MDCK cells in 6 well plates were inoculated with virus stock for 1 hr, then an overlay (mixture of equal volume of DMEM and 2.4% Avicel (Sigma-Aldrich, UK) supplemented with 1 µg/ml TPCK-treated trypsin and 0.14% BSA fraction V) was put onto the wells. After 48 hr, cells were fixed using 3.5% formaldehyde and stained with 0.1% crystal violet. Virus titres were calculated by plaque count*dilution factor/(volume of inoculum) and expressed as PFU/ml. Plaque sizes were quantified by ImageJ software. HA assays were performed as previously described (*Hutchinson et al., 2008*). To analyse virion composition, clarified allantoic fluid was pelleted through a 33% sucrose pad at 91,000 *g* as previously described (*Hutchinson et al., 2008*).

## Reverse transcription quantitative polymerase chain reaction (RT-qPCR)

For quantification of vRNA in culture supernatants, RNA was extracted from viral supernatants using QIAamp Viral RNA Mini Kits (Qiagen, Manchester, UK). DNA was removed by either RNase-Free DNase Set (Qiagen, Manchester, UK) or RQ1 RNase-Free DNase (Promega, Southampton, UK). Reverse transcription quantitative polymerase chain reactions (RT-qPCR) were carried out using Bioline SensiFAST one-step RT-PCR kit (BIO-82020) with modified cycling conditions – 45°C for 10 min, 95°C for 2 min, then 40 cycles of 95°C for 10 s, then 60°C for 30 s using segment 1 and 5 primers listed in *Table 2*).

## Replication phenotype in vitro

MDCK cells were infected in triplicate with WT virus and the mutants at a multiplicity of infection (MOI) of 0.01 (multi-step replication cycle) or 1 (single replication cycle). The supernatants were

Table 2. Primers used for qPCR and for amplification of segment 5 for competition assays.

| Gene/Region | Aim | Primer type | Sequence |
|---|---|---|---|
| Seg 5 | Reverse Transcription | Sense | ATCATGGCGTCTCAAGGCAC |
| Seg 5 | Sequencing | Sense | GAATGCCACTGAAATCAGAG |
| | | Antisense | CGTCCGAGAGCTCGAAGACT |
| Seg 5 | Competition Assay | Sense | CCAGAATGCCACTGAAATCA |
| | | Antisense | CCTTGCATYAGMGAGCACAT |
| Seg 5 | RT-qPCR | Sense | ATCATGGCGTCTCAAGGCAC |
| | | Antisense | CCGACGGATGCTCTGATTTC |
| GAPDH | RT-qPCR | Sense | CTACCCCCAATGTGTCCGTCG |
| | | Antisense | GATGCCTGCTTCACCACCTTC |
| IFN-β | RT-qPCR | Sense | CACAGCCCTCTCCATCAACT |
| | | Antisense | GCATCTTCTCCGTCATCTCC |

harvested at multiple time points (12, 24, 48 hr post-infection). Viral titres were similarly determined at 24 hr using 5 replicate cultures. RNA/infectivity ratios were determined by determining RNA amount in viral supernatants by qPCR using primers from segment 5 and comparing this with their infectivity titres.

## Plating efficiency

A549 cells were infected at an empirically determined MOI (by titration) that gave ~10% levels of infection for 6 hr, after which cells were fixed, permeablised and stained for a range of viral proteins. M2 was detected by ab5416 (Abcam, Cambridge, UK), NS1 by an in-house (rabbit) antiserum, NA by an in house (mouse) antiserum, NP by ab20343 (Abcam, Cambridge, UK) and PB2 by an in house rabbit antiserum. Secondary antibodies were species specific Alexafluor-488 or Alexafluor-546 (A21202, A11005, A21207 and A21206, Life Technologies, Loughborough, UK).

## Amplification of IAV stocks in embryonated eggs

200PFU of virus stocks were inoculated into 11 day old chick eggs and incubated for 2 days. Eggs were chilled at 4°C overnight, then allantoic fluid was harvested, checked for virus by HA assay, and pooled as virus stock. Stocks were centrifuged and then ultracentrifuged at 48,000g for 90 min on a 30% sucrose cushion. Pellets were resuspended directly in laemmli buffer for SDS-PAGE and coomassie staining following standard protocols.

## Competition assays

To compare the relative replication fitness of WT virus and the mutants, equal titres of a pair of viruses were mixed as the 'starting inoculum' to infect A549 cells simultaneously. Following development of a cytopathic effect (CPE), the supernatant was diluted by hundred fold and used to inoculate fresh A549 cells for the next passage. After 5 or 10 passages, the supernatant was withdrawn for RNA extraction and reverse transcription. A specific region on Segment 5 was amplified by a primer pair that anneals equally to the 2 mutants (*Table 2*). The amplicons were then treated with restriction enzymes that only digest one of the 2 amplicons (*Table 3*). Similar procedures were also performed on the starting inoculum. The digested PCR products were separated by electrophoresis. The ratios of digested and undigested bands were quantified and normalized by the ratio in the starting inoculum.

## Replication phenotype in vivo

This was assessed in the BALB/c mouse model. All animal experiments were carried out under the authority of a UK Home Office Project Licence (60/4479) within the terms and conditions of the strict regulations of the UK Home Office 'Animals (scientific procedures) Act 1986' and the Code of Practice for the housing and care of animals bred, supplied or used for scientific purposes. Groups of six

8 week-old female BALB/c mice were intranasally infected with between 20 and 200 PFU of influenza virus PR8 strain in 40 μl DMEM under oxygenated isofluorane anaesthesia. Mice were weighed daily and assessed for clinical signs of infection. At various time points post-inoculation, mice were euthanized and whole blood was collected with subsequent collection of plasma supernatant after coagulation (for microneutralisation assays). Left lungs were harvested for qPCR (as above), plaque assay (as above), mouse cytokine arrays (below) or T cell assays (below). The four lobes of the right lung were inflated with and then immersed in 10% neutral buffered formalin (Sigma-Aldrich, UKSigma-Aldrich) until fixed, then processed using routine methods and embedded in paraffin blocks. 5 μm thin section were cut and stained with haematoxylin and eosin for histological examination. The sections were assessed (blinded) by a veterinary pathologist (PMB). The individual pathology features assessed were damage to the airway epithelium (degeneration, necrosis and repair), perivascular inflammation, peribronchi/bronchiolar inflammation, interstitial inflammation, interstitial necrosis and type II pneumocyte hyperplasia. Each feature was scored from 1 (mild) to 3 (marked). The percentage of lung affected was also noted. A challenge experiment was undertaken, whereby mice in groups of 4 were inoculated with 100, 50, or 20 PFU of CpG-high virus, or mock infected (DMEM only). At 3 weeks post-inoculation, mice were challenged with 200 PFU of WT PR8 virus. Mice were euthanised at 6 days post-challenge.

## Mouse cytokine array

Equal volume of homogenates of lungs in the same group was pooled together. Levels of cytokines were determined using Mouse Cytokine Antibody Array - Panel A (R&D systems ARY006) commercial kit according to its instruction. Briefly, cytokines in lung homogenates were detected by a detection antibody cocktail and subsequently attached to antibody spots printed on membranes with each spot recruiting one cytokine. A fluorescence dye was then added and fluorescence of each spot was visualised and quantified using LI-COR Odyssey Infrared Imaging System (LI-COR, Cambridge, UK). For each cytokine, spots were printed in duplicate. An average value was calculated and normalized by the mock-infected control. A heat map was generated based on the folds of induction over the mock-infected group. Cytokines with >two fold induction over mock-infected group and with >two fold induction or inhibition compared with the WT-infected group were highlighted in a parallel heat map.

## Microneutralisation assay

Mouse sera were pre-treated with receptor destroying enzyme using the RDE Kit (Cosmos Biomedical, Swadlincote, UK) according to manufacturer's instructions. Confluent MDCK cells in 96 well plates were washed to remove FCS, and mouse sera were added to plates and two-fold serially diluted down the plate to a final dilution of 1:1280. 200 PFU/ well of PR8 virus stock was added, and wells were overlaid 1:1 with Avicel containing trypsin and BSA as described previously. Cells were incubated at 37°C overnight. Overlay was removed and the cells were fixed with 3.5% paraformaldehyde and permeabilised with 0.2% Triton-X100. Primary antibody reactive for influenza A NP was added 1/1000 (Abcam, Cambridge, UK, ab20343) in ELISA buffer (10% horse serum, 0.1% Tween-20) and incubated for 1 hr. Cells were washed and secondary antibody (peroxidase labelled anti-mouse antibody, Thermo scientific, Loughborough, UK SA1-100) was added 1/1000 in ELISA buffer

**Table 3.** List of enzymes used in pairwise competition assays.

| Virus 1* | Virus 2* | Restriction Enzyme | Digestion Site |
|----------|----------|---------------------|----------------|
| WT | CDLR | Hpy188III | In CDLR |
| WT | CpGH | Hpy188III | In CpGH |
| WT | UpAH | Hpy188III | In UpAH |
| CpGH | UpAH | AhdI | In CpGH |
| CpGH | CDLR | AhdI | In CpGH |
| UpAH | CDLR | BsaHI | In UpAH |

*Virus pairs to be differentiated

for 1 hr. Cells were washed and treated with True Blue Peroxidase (KPL, 50-78-02) according to manufacturer's instructions, and then visually inspected for colorimetric differences.

## Intracellular cytokine staining (ICS) and flow cytometry

Mice were euthanized at day 21 post-inoculation and the lungs and spleen were removed for analysis. Single cell suspensions were generated from perfused lung by incubating the dissociated tissue in RPMI containing DNase and collagenase, and then passing the digested tissue through a 45 µm cell sieve to release the tissue-bound lymphocytes. These were treated with ACK lysing buffer (Life Technologies, Loughborough, UK) to remove red blood cells and then resuspended in RPMI+10% FCS. Spleens were dissociated through a 45 µm cell sieve and then ACK lysing buffer was used to remove red blood cells. The splenocytes were resuspended in RPMI+10% FCS. Lung and spleen single cell suspensions were counted using an automated cell counter (Biorad). $10^6$ splenocytes or lung lymphocytes were incubated for 6 hr at 37°C with a cocktail of peptides (NP$_{147-155}$ (TYQRTRALV), HA$_{518-526}$ (IYSTVASSL), HA$_{462-472}$ (LYEKVKSQL) and HA$_{126-138}$ (HNTNGVTAACSHE) (ProImmune, Oxford, UK)) designed to activate both CD8+ and CD4+ influenza PR8-specific T cells. Media-only wells were included as negative controls. Golgi-stop was added 2 hr post-stimulation and stimulation was stopped by storing the cells at 4°C 4 hr later. ICS was performed according to methods provided with the Cytofix/Cytoperm kit (BD Bioscience, Oxford, UK). Briefly, the cells were treated with Fc block (eBiosciences, Hatfield, UK) and then labelled with Fixable Near-IR Dead Cell Stain Kit (Life Technologies, Loughborough, UK) and the fluorochrome conjugated monoclonal antibodies CD8-PerCPCy5.5 (eBiosciences, Hatfield, UK) and CD4-eFluor450 (eBioscience), followed by fixation and permeabilisation. The anti-cytokine antibodies IFN-γ-APC (eBiosciences, Hatfield, UK), TNF-α-FITC (eBiosciences, Hatfield, UK) and IL-2-PE (BD Bioscience, Oxford, UK) were added, and then cells were washed and resuspended in FACS buffer (PBS, 2% FCS, 0.1% NaN$_3$, 5 mM EDTA). Samples were analyzed on the LSRII flow cytometer (BD Biosciences, Oxford, UK).

## IFN-γ ELISpot assay

IAV peptides were selected to match the sequence of the PR8 strain of IAV and to represent immunodominant epitopes in the BALB/c strain (Influenza Research Database, http://www.fludb.org). CD8 + T cells from BALB/c mice respond strongly to an epitope in the NP protein NP$_{147-155}$ (TYQRTRALV) (NP) and two further epitopes in the HA protein, HA$_{518-526}$ (IYSTVASSL) (HA-dom) which is dominant and HA$_{462-472}$ (LYEKVKSQL) (HA-sub) which induces weaker responses. The HA protein also contains a CD4+ T cell epitope HA$_{126-138}$ (HNTNGVTAACSHE) (HA-CD4), which was measured in these experiments.

IFN-γ ELISpot assays were performed on frozen splenocytes thawed and rested overnight in 3 ml RPMI+10% FCS (R10) in 37°C, 5% CO$_2$. Assays were performed according to the manufacturer's instructions (Mabtech, Sweden). Briefly, rested splenocytes were plated in duplicate at a concentration of $1–2 \times 10^5$ splenocytes per well in Hydrophobic Immobilon Membrane plates (Merck Millipore, Watford, UK) coated with 100 µl of anti-IFN-γ (10 µg/ml) (Mabtech, Sweden). Cells were stimulated with the individual peptides (3 µg/ml final concentration) which constituted the peptide cocktail used in the ICS. Cells were stimulated with R10+0.1% DMSO as negative control. After overnight stimulation at 37°C, 5% CO$_2$, the wells were washed and the number of IFN-γ producing cells detected by addition of biotinylated anti-IFN-γ (Mabtech) (50 µl at 1 µg/ml), followed by Streptavidin-AP (Vectorlabs, Peterborough, UK) (50 µl at 1:750 dilution) and the substrate NBT/BCIP (Thermo Fisher, Loughborough, UK). The number of spots generated in each well was counted using an automated ELISPOT reader (AID Elispot, Wheatley, UK). The number of spots generated in the negative control wells was used to determine the limit of detection of the assay (5 spots per $10^6$ cells).

## Acknowledgements

We are grateful to staff at the animal handling facility for assistance with the in vivo studies and Matthew Turnbull for providing cell lines.

## Additional information

### Funding

| Funder | Grant reference number | Author |
|---|---|---|
| Wellcome Trust | WT103767MA | Eleanor Gaunt<br>Huayu Zhang<br>Nicky J Atkinson<br>Peter Simmonds |
| Biotechnology and Biological Sciences Research Council | BB/J004324/1 | Helen M Wise<br>Marlynne Quigg Nicol<br>Philippa M Beard<br>Bernadette M Dutia<br>Paul Digard<br>Peter Simmonds |
| Oxford Martin School, University of Oxford | WT091663MA | Lian N Lee<br>Andrew J Highton<br>Paul Klenerman |
| Medical Research Council | MR/K000276/1 | Paul Digard |

The funders had no role in study design, data collection and interpretation, or the decision to submit the work for publication.

### Author contributions

EG, HMW, HZ, LNL, NJA, Acquisition of data, Analysis and interpretation of data, Drafting or revising the article; MQN, AJH, Acquisition of data, Drafting or revising the article; PK, PMB, BMD, Analysis and interpretation of data, Drafting or revising the article; PD, PS, Conception and design, Analysis and interpretation of data, Drafting or revising the article

### Author ORCIDs

Peter Simmonds, http://orcid.org/0000-0002-7964-4700

### Ethics

Animal experimentation: All animal experiments were carried out under the authority of a UK Home Office Project Licence (60/4479) within the terms and conditions of the strict regulations of the UK Home Office 'Animals (scientific procedures) Act 1986' and the Code of Practice for the housing and care of animals bred, supplied or used for scientific purposes.

## Additional files

### Supplementary files

• Supplementary file 1. Sequences of the mutated regions in segment 5 of IAV (CDLR, CpG-high and UpA-high) are provided in the Supplementary file.

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
