## [Decision Letter]

Thank you for submitting your work entitled "Elevation of CpG frequencies in influenza A genome attenuates pathogenicity but enhances host response to infection" for consideration by *eLife*. Your article has been reviewed by three peer reviewers, one of whom, Marc Lipsitch, is a member of our Board of Reviewing Editors, and the evaluation has been overseen by Richard Losick as the Senior Editor.

The reviewers have discussed the reviews with one another and the Reviewing Editor has drafted this decision to help you prepare a revised submission. For reasons stated below we consider this an important Research Advance related to your prior *eLife* publication.

Summary:

This paper describes the attenuation of influenza viruses by altering the frequency of CpG and UpA dinucleotides. As way of background, there have been a number of papers arguing that viruses can be attenuated by altering their "codon-pair" frequencies. Some of the authors of this paper have argued that this codon-pair de-optimization was in fact due to the fact that altering codon pairs also increased CpG and UpA dinucleotides. They have published prior work showing that altering the dinucleotide frequencies without altering codon pairs leads to similar levels of attenuation. Based on this work, they have argued that CpG/UpA di-nucleotides mediate attenuation of mammalian viruses though some as-yet unknown mechanism. This study is an important continuation of previous work from this group exploring the biology of codon bias in virus-host interactions using echovirus 7. The paper is lucid and experiments performed at the highest levels of technical competence. The findings will be of general interest for both basic and translational scientists.

In the current work, the authors generate influenza viruses where they alter the CpG/UpA content in influenza NP. They examine four viruses: wildtype PR8, PR8 with scrambled nucleotides but no systematic change in dinucleotides, viruses with more CpG, and viruses with more UpA. They find that the CpG and UpA viruses are greatly attenuated both in MDCK cells and mice. The scrambled control appears slightly attenuated in MDCK cells and not detectably attenuated in mice. The CpG/UpA viruses induce substantial levels of cytokines despite replicating less well. This is used to note that such viruses might be a good way to create vaccines.

Reasons for converting the article to a Research Advance:

Overall, all reviewers found the article interesting but felt it did not demonstrate its central argument – that dinucleotide frequencies are the mechanism of viral attenuation, with sufficient certainty to merit publication in *eLife*. The primary problem is as follows: there are undoubtedly a great number of ways that synonymous mutations can alter viral fitness, including altering with RNA structure, altering translation efficiency, altering genome packaging efficiency, introducing spurious splicing, etc. So how are we to know that the CpG/UpA altered viruses are really attenuated *because* of their altered di-nucleotide content, rather than due to some other effect of one of the numerous synonymous mutations. We are supposed to be re-assured that this is not the case because the scrambled control. While such a scrambled control is necessary to make such a case (and certainly should have been included in all of the now dubious codon-pair de-optimization studies), it is not sufficient. In effect, we have one data point for the control (the scrambled control) and two data points for the di-nucleotide alteration.

Put another way, every nucleotide sequence can be characterized by a number of summary statistics – nt frequency, nt pair frequency, codon frequency, codon pair frequency, codon triplet frequency, RNA folding energy, etc. These often multivariate summary statistics (e.g. codon pair frequency, which contains values for each pair) can be further summarized down to a single summary – such as CPS. Each of these summary statistics is correlated with many of the others. It is certainly true that IF summary statistic A is mechanistically causal, and summary statistic B is not, and there are no other important features of the sequence controlling attenuation *then* creating variant sequences that change A but hold B constant should attenuate, while variants that change B without changing A should not attenuate. The converse doesn't follow. The true cause of attenuation could be some feature that no one has thought to look for, something perhaps as simple as the frequency of one particular codon that tends to change when adapting sequences to measure A but not measure B. Or it could be something extremely complex involving long-range interactions that for some reason tend to be much more sensitive to measure A than to measure B. Just as the recognition that changing CPS tends to change CpG frequencies generated an alternate hypothesis for attenuation, one could generate other alternate hypotheses for summaries that correlate with these measures. For practicable numbers of experiments, it will be hard to exclude – based on the attenuation of certain sequences but not others – all but one hypothesis for the causal mechanism.

Given the inability to sample all possible properties of a sequence that might attenuate a virus, we believe the only way to establish the scientific point that there is a specific mechanism to inhibit viruses with altered di-nucleotides would be to isolate the host factor(s) that are specifically responsible for the CpG/UpA attenuation, and show that there is a specific biophysical interaction triggered by CpG/UpA dinucleotides. By analogy, the finding that a particular sequence is the mechanism of bacteriophage restriction is made far more convincing by the discovery of a specific restriction enzyme with the specified target sequence that restricts phage replication and that, when knocked out in the bacterial host, no longer does so.

It is not reasonable in the scope of an *eLife* revision to request such a further discovery. However, in the absence of such data, this study adds further circumstantial evidence to the authors' prior work suggesting di-nucleotides can attenuate viruses. But on its own, the data here don't fully clinch the case and certainly don't provide a sufficient advance over the prior work to merit publication as a new Research Article in *eLife*.

The experimental evidence provided by this work, though not conclusive, is an advance over the prior work. Moreover, another interesting advance is thatthe attenuated di-nucleotide viruses are also attenuated in vivo, and induce good immunity. While not terribly surprising this is potentially useful from a practical perspective and would further justify publication as a Research Advance.

Essential revisions:

1) The reviewers were concerned about the comparisons of wt vs. mutated viruses based solely on PFU. As recently reported by a number of labs, flu virus preps can differ widely in their number of semi-infectious particles, and that in one case, this was modulated by sequence alterations. It would be nice to see that the different viruses don't vary widely in their particle to PFU ratios. This can be easily done by flow cytometry at very low MOI to enumerate the number of particles capable of producing viral proteins following infection. For example see http://jvi.asm.org/content/87/6/3155.short. At the same time, the findings can be corroborated just based on HA titrations: which despite its simplicity is perhaps the most accurate method for counting particles (at the end point, there is one particle for every two RBC).

2) The flow analysis using mAbs for HA, NA, NS1, M1, M2, and NP to stain cells infected at very low MOI would also demonstrate whether to what extent mutants synthesize major viral proteins at the same rate as wt virus from the same number of input particles (one particle per cells).

3) It is well established that UV or γ-irradiated virus, despite not replicating at all, is highly immunogenic for T cell responses, and similarly less immunogenic for antibody responses. This should be mentioned and incorporated into the Discussion, viz., what are the advantages/disadvantages of dinucleotide modified viruses vs. inactivated virus.

---

## [Author Response]

In response to the requested essential revisions, we have made the following changes:

1) We have performed several further investigations of the infectivity of WT and mutant version of IAV quantified by HA assay (new Figure 1). The principal findings from these additional experiments (as summarised in the second paragraph of the subsection “Replication phenotypes of segment 5 CpG- and UpA-high mutants”) was that there was indeed a consistent 3-5 fold increase in virus particle (quantified by either HA or qPCR assays) to infectivity ratios of CpG-high and UpA-high mutants compared to WT (or CDLR permuted control) variants. We additionally showed that this difference was not the result of a packaging defect since:

A) Immunostaining of cells infected at low multiplicity for a range of IAV proteins from 5 of the 8 segments at 6 hours post-infection revealed comparable proportions of cells positive for each protein between WT and mutant forms of IAV (Figure 1—figure supplement 3).

B) The relative amounts of NP, HA1, HA2 and M viral structural proteins in partially-purified virions from the different mutants were comparable by PAGE analysis (Figure 1—figure supplement 4).

C) Quantitation of segment 5 and 2 RNAs in virus stocks indicated little variation in the relative proportions of each segment between WT and IAV mutants (Figure 1—figure supplement 5).

These new results that were requested by the reviewers are summarised in the Results section in the revised manuscript. We have not attempted to estimate particle numbers from the HA titres as there is unfortunately no universal conversion factor to go from HAU to particle number – binding is affected by many other factors such as HA affinity and specificity for sialic acid, as well as NA activity/specificity and particle shape. However, the relevant analysis is the ratio between HA titre and infectivity and the potential existence of differences in this ratio between WT and mutant strain of IAV.

Collectively, the additional experiments have provided no evidence for a packaging defect in virions of CpG- and UpA-high mutants, yet has confirmed differences in virion / infectivity ratios. As a more general perspective, increasing CpG and UpA frequencies in other virus systems (e.g.echovirus 7) also increases RNA/particle to infectivity ratios. E7 is a monopartite virus indicating that alterations to RNA / infectivity ratio as a result of dinucleotide frequency changes occur independently of packaging. Summarily, the reviewers requested further analysis in the form of HA assay, which we have undertaken and supported by generating HA:infectivity and RNA:infectivity ratios.

2) To demonstrate that the rate and of protein synthesis was the same for wildtype and mutated viruses, we (as described in the previous section) have quantified by immunofluorescence the numbers of cells expressing a range of viral proteins: NP, M2, NS1, NA and PB2. Cells produced these proteins at approximately the same rate (Figure 1—figure supplement 3). To show that the same *amount* of protein is produced, virus stocks were analysed by PAGE, and relative ratios of the different virus structural proteins were quantified. The relative amounts of protein production were consistent, demonstrating that NP production was not defective in our mutant viruses (Figure 1—figure supplement 4). Thus, we have not observed any evidence for reduced packaging of individual segments in the CpG and UpA-high virus mutants (Figure 1—figure supplement 3, 4 and 5).

3) To meet the guidelines of a Research Advance, we have reduced the word count of the study and it is problematic to include a further discussion of vaccine strategies in the specific context of the paper. We have however edited the section describing different strategies for IAV attenuation to include reference to inactivated vaccines and their ability to induce protective T cell responses (subsection “Development of an attenuated vaccine for influenza A virus”, second paragraph).